# Influence of Laser Beam Wobbling Parameters on Microstructure and Properties of 316L Stainless Steel Multi Passed Repaired Parts

**DOI:** 10.3390/ma15030722

**Published:** 2022-01-18

**Authors:** Artem Aleksandrovich Voropaev, Vladimir Georgievich Protsenko, Dmitriy Andreevich Anufriyev, Mikhail Valerievich Kuznetsov, Aleksey Alekseevich Mukhin, Maksim Nikolaevich Sviridenko, Sergey Vyacheslavovich Kuryntsev

**Affiliations:** 1Institute of Laser and Welding Technologies, Saint Petersburg State Marine Technical University, Lotsmanskaya Str. 3, 190121 Saint Petersburg, Russia; t-voropaev94@mail.ru (A.A.V.); vova.protsenko1996@mail.ru (V.G.P.); dmitriyanufriyev23@yandex.ru (D.A.A.); 2N.A.Dollezhal Research and Development Institute of Power Engineering, Malaya Krasnoselskaya Str. 2/8, 107140 Moscow, Russia; a.muhin@nikiet.ru (A.A.M.); sviridenko@nikiet.ru (M.N.S.); 3Department of Materials Science and Welding, Kazan National Research Technical University Named after A.N. Tupolev—KAI (KNRTU—KAI), 420111 Kazan, Russia

**Keywords:** laser welding, 316L austenitic stainless steel, repair, microstructure, mechanical properties, cold filler material

## Abstract

The results of experimental studies of repair of the supporting structure components made of 316L steel multi-pass laser cladding with filler wire are presented. The influence of the wobbling mode parameters, welding speed, and laser power on the formation of the deposited metal during multi-pass laser cladding with filler wire of 316L steel samples into a narrow slot groove, 6 mm deep and 3 mm wide, are shown. Non-destructive testing, metallographic studies, and mechanical tests of the deposited metal before and after heat treatment (2 h at 450 °C) were carried out. Based on the results of experimental studies, the optimal modes of laser beam wobbling were selected (amplitude—1.3 mm, frequency—100 Hz) at which the formation of a bead of optimal dimensions (height—1672 μm, width—3939 μm, depth of penetration into the substrate—776 μm) was ensured. A laser cladding technology with ESAB OK Autrode 316L filler wire has been developed, which has successfully passed the certification for conformity with the ISO 15614-11 standard. Studies of the chemical elements’ distribution before and after heat treatment showed that, after heat treatment along the grain boundaries, particles with a significantly higher Mo content (5.50%) were found in the sample, presumably precipitated phases. Microstructure studies and microhardness measurements showed that the upper part metal of the third pass, which has a lower microhardness (75% of base metal), higher ferrite content, and differently oriented dendritic austenite, significantly differs from the rest of the cladded metal.

## 1. Introduction

Austenitic stainless steels are a very widespread class of structural materials used in various industries. 316L SS is cryogenic structural steel with low carbon content, making it less likely to form chromium carbides. Of all steels of the 300 series, steel 316L is the most resistant to all types of corrosion, including pitting corrosion, as it is alloyed with molybdenum (2–3%), except for nitric acid, which is a strong oxidizing agent for this steel. As steel 316L has almost three times less carbon than steel 316, it has better weldability. Additionally, this steel has high strength, elasticity, and ductility, a good set of technological properties. The main disadvantage of this steel is its high tendency to undergo stress corrosion cracking, which, among other things, can be caused by the thermal effect of the welding cycle with a high level of heat input.

The phase composition of the weld plays an important role and has a significant effect on the properties of welded joints obtained by welding 300 series steels. The phase composition of the weld depends on the chemical composition of the base metal, the chemical composition of the filler material [1,2], (*Cr_eq_/Ni_eq_*) and cooling rates, especially in some critical temperature ranges [3]. In turn, the cooling rates depend on the selected type of welding [4], welding modes [5,6], and the presence of concurrent heating [3,4,7] or cooling during welding or recovery cladding [8]. Additionally, the phase composition can vary in depth of the weld seam (top, middle, and bottom parts of the seam), for example, when welding with a defocused laser beam or laser beam using the longitudinal wobbling mode [9,10,11]. That also has a significant influence on the performance properties of products. The cooling rates also play a significant role in the formation of various forms of ferrite (delta, sigma, and X) [4,12] and the formation of unfavorable phases (carbides, nitrides, and different intermetallic compounds), in particular the formation of chromium nitrides, as nitrogen poorly dissolves in δ-ferrite.

The content of δ-ferrite (5–15%) can lead to a decrease in the formation of solidification cracks due to the different thermal expansion coefficients of ferrite and austenite (α_γ_ = 17.3, α_δ_ = 9.9 × 10^−6^ m/mK). Additionally, such harmful impurities as sulfur and phosphorus dissolve well in δ-ferrite. The cooling rate in the range of 1200–800 °C significantly affects the amount of formed δ-ferrite.

In particular, the performance properties of welded joints made of austenitic corrosion-resistant steels can be affected by nitrogen content in the shielding gas, the level of heat input, interpass temperature range, and others.

According to formulas (1) and (2), for this steel, *Cr_eq_/Ni_eq_* = 1.62; that is, this steel is resistant to hot cracks taking into account the content of S + P = 0.075.
(1)$$Cr_{eq}\;=\;Cr(17)+1.37Mo(2.5)+1.5Si(0.75)+2Nb+3Ti$$
(2)$$Ni_{eq}=Ni\;\left(12\right)+0.31Mn\;\left(2\right)+22C\;\left(0.03\right)+14.2N+Cu$$

For 316L steel, the solidification mode will be *Cr_eq_/Ni_eq_* dependent. According to the identity 1.48 < *Cr_eq_/Ni_eq_* < 1.95, the solidification mode is supposed to be ferritic-austenitic (FA):(3)$$L\to L+\delta \to L+\delta +{\left(\gamma +\delta \right)}_{per/eut}\to \gamma +\delta$$

However, in this case, one should take into account the high cooling rates of the weld seam metal, which can lead to the formation of a higher amount of *δ*-ferrite. If it is planned to operate the product at high temperatures, it is necessary to carry out heat treatment after fusion welding to reduce the amount of *δ*-ferrite.

Currently, welding and repair of large thickness products made of steel 316L are most relevant for the nuclear industry, the military complex, shipbuilding, and the pipe industry. Most enterprises in these industries use arc-welding methods to make welded joints. However, the use of traditional technologies is limited by several disadvantages of an electric arc heat source. A high level of specific heat input, low welding speed, as well as a large number of passes during welding with large thicknesses lead to the appearance of welding deformations. Significant overheating of the product’s metal after each pass increases the time of the technological process due to the increase in the time for the interpass cooling temperature, which for steels of the 300 series is 150–170 °C. The wide groove preparation used before arc welding also leads to an increase in the time for preparatory operations and the loss of expensive metal.

Beam welding technologies, such as laser welding [13,14] and electron beam welding [15], have proved satisfactory when welding large thickness parts. However, the high cost of equipment and stringent requirements for the processed surfaces do not allow the use of these methods everywhere. Therefore, there are various combined methods such as hybrid laser-arc welding [16] and laser welding with cold filler wire [17,18,19,20].

The main problem for obtaining a high-quality welded joint at laser welding using a welding wire is the positioning accuracy of the laser beam relative to the cold filler material [21,22]. The diameter of the focused laser beam in the welding zone is extremely small, 100–300 µm, while the wire diameter is usually at least 1 mm. The wire can go beyond the range of the laser beam action due to uneven feed during the welding process and its rigidity, thereby forming an uneven bead-on-plate [23,24,25,26].

Under the conditions of multi-pass welding, these irregularities can result in extensive zones of lack of fusion [27,28]. When the laser spot diameter is increased by defocusing due to the Gaussian distribution of energy in the spot, there is a loss of power at the edges [9,10]. In this case, a further increase in the laser irradiation power is necessary. An increase in the laser irradiation power can lead to a transition from convective melting to the formation of a key-hole [10,29], the behavior of which is less stable and can lead to the formation of pores in the fusion zone and mixing of two layers or the base metal. In addition, an increase in power can lead to undesirable overheating of the near-weld area.

An alternative to a defocused laser beam is the use of a longitudinal, transverse, or circular wobbling of the beam (laser or electron), which is widely used in both laser and electron beam technologies [30,31]. Moreover, optimization of the beam wobbling parameters can lead to the minimization of defects and unevenness of the weld bead.

Welding deformations are an additional problem in multi-pass welding. When performing a large number of passes, the metal is repeatedly heated. With each subsequent pass, the welding deformations increase. Reducing the number of passes by using special grooves can reduce the negative effect of overheating on welding deformations [10,20,32,33].

Modern high-power lasers allow welding with a thickness of about 10 mm at speeds up to 30–60 mm/s [34] however, at such speeds, it will be necessary to use a wire of a larger diameter, which will be even more rigid, and high crystallization rates will not allow alloying elements to be uniformly distributed over the entire depth of the seam [35].

It follows from the above that the use of cold filler material laser welding compared to arc or laser-arc welding or recovery cladding will significantly increase the productivity of repair works due to the high cooling rates of each pass, a decrease in the time delay between each subsequent pass, and a decrease in heat input into the component being repaired [36,37,38]. Moreover, the optimization of laser beam wobbling parameters in comparison with defocusing of the laser beam will lead to the formation of a defect-free groove filling during repair and the use of lower laser irradiation powers, which is energetically more efficient.

The aim of this work is to study the effect of laser irradiation power, welding speed, laser beam wobbling amplitude, cold filler wire feed rate on the quality of bead-on-plate formation, microstructure, and distribution of chemical elements depending on heat treatment at laser multipass recovery cladding of 316L steel filler wire, an ESAB OK Autrode. The development of technology and technological process for laser recovery cladding with cold filler wire, which will ensure the formation of deposited metal that meets the requirements of quality level B according to ISO 13919, is another aim.

## 2. Experimental

### 2.1. Planning of Experiment and Equipment

Within the framework of the experimental study, we varied such parameters of laser recovery cladding as the laser irradiation power, welding speed, cold filler wire feed rate, and the amplitude of laser beam wobbling in the plane perpendicular to the process direction. A one-factor experiment was carried out to assess the influence of each of the mode parameters on the studied characteristics.

Experimental studies were carried out in the laboratory of laser and additive technologies of the Institute of Laser and Welding Technologies—ILWT, St. Petersburg State Marine Technical University using a technological laser complex of the portal type. The appearance of the complex is shown in Figure 1. Table 1 shows the technical characteristics of the complex. The complex is made based on an LS-15 ytterbium fiber laser (IPG-Photonics, Fryazino, Russia) with a maximum output power of 15 kW, linear guides (Isel), a Precitec YW 52 laser head with focal lengths of focusing and collimating lenses of 385 mm and 150 mm, respectively, rigidly fixed on the carriage and moved along the linear guides of the portal. Laser irradiation was fed to the laser head through a transport fiber of 200 microns in diameter. A PDGO-510 feeding mechanism (ITS, Simferopol, Russia) was used to feed the cold filler wire.

Laser cladding with cold filler wire was carried out according to the scheme shown in Figure 2. The filler wire was fed in front of the laser beam. The distance L between the axis of the laser beam and the axis of the wire was half the radius of the pilot spot when the focal plane was deepened by 12 mm. At this depth, the spot diameter was 2.2 mm the distance L was 0.5–0.6 mm, respectively. The inclination angle of the optical head was 5 degrees. The inclination angle of the filler wire feed was 30 degrees. The shielding gas was supplied through 4 copper tubes located behind the laser beam. The total shielding gas flow rate was 20 L/min.

### 2.2. Bead-on-Plate Test and Cold Filler Laser Recovery Cladding

316L SS, compared to 316 SS, has better weldability due to its lower carbon content. As a rule, the phase composition of the weld seam metal or clad metal depends on the filler material and cooling rates [10], preheating temperature [39], and interpass distance [40]. According to formula (3), the solidification mode will be ferritic-austenitic; however, as presented by the authors of [10], during multi-pass welding with a cold filler wire, solidifications cracks may form in the first passes developing along the primary austenitic solidified island located at the weld centerline, although the bulk of the weld metal is solidified in primary ferritic mode. Depending on the preheating temperature, the deposited metal may have a columnar microstructure at its interface with the substrate and an equiaxial microstructure in the center of the weld seam.

Table 2 presents the chemical composition of the base metal and filler material. The filler material was ESAB OK Autrode 316L wire, 1 mm in diameter.

Before multipass cladding, one bead was deposited on a flat substrate to identify the optimal parameters of laser process with a cold filler wire. The main varied parameters were the laser irradiation power, welding speed, filler wire feed rate, and the amplitude of laser beam wobbling are given in Table 3. Table 4 gives the parameters of the laser cladding mode with cold filler wire with the highest quality of the bead-on-plate test.

Such a low welding speed and power were chosen to avoid the formation of a key-hole welding, stable melting of the filler material, and minimize cooling rates.

Samples with overall dimensions of 200 × 100 × 10 mm for testing the technology of laser welding with a cold filler wire and samples with overall dimensions of 450 × 150 × 10 mm for certification of the technology were made of 316L steel by plasma cutting. The groove and welded edges were milled. Figure 3 shows the samples prepared to simulate the repair process of a defect.

Before welding, the surfaces mating to the edge to be welded were treated to a metallic luster using an angle grinder (Institute of Laser and Welding Technologies, Saint Petersburg State Marine Technical University, Saint-Petersburg, Russia) and then degreased with acetone. The tack welds were placed on the reverse side of the welded surface with a distance between the tack welds of 100 mm and 50 mm from the end of the plate.

Before multi-pass cladding, the sheets were tack welded with four tack welds, 2 per side. The distance between the tack welds was 100 mm. The tack welded sample was installed in special tooling to reduce welding deformations. After cladding, the sample was heat-treated in a fixed position on the tooling. The interpass distance along the *Z*-axis was 3 mm.

The inert gas argon of the highest grade was used as a shielding gas under the requirements of the standard ISO 14175-I1 with a flow rate from the front side of 20 L/min.

### 2.3. Microstructure, Microhardness, Corrosion Tests, and Non-Destructive Control

The Department of Materials Research, ILWT (Institute of Laser and Welding Technologies, Saint Petersburg State Marine Technical University, Saint-Petersburg, Russia) produced the macrosections and studied the quality of formation of the cladded layer. Macrosections were made using grinding equipment (Buhler, Lake Bluff, IL, USA). The samples were etched with a solution of HCl with HNO_3_ in a ratio of 3:1.

The microhardness of the cladded layers was measured at room temperature on a Micromet 5103 microhardness tester (Buhler, Lake Bluff, IL, USA), at a load of 500 g according to DIN EN ISO 14577.

The quality assessment of welded joint formation was carried out using optical and visual-measuring methods of control. The optical method was implemented using a Leica DMi8 A optical microscope (Synercon, Moscow, Russia) according to BS EN 61326-1: 2013. Visual-measuring control (BS EN ISO 23277: 2015) was carried out under the requirements of the ISO 13919-1 standard (quality level B). The microstructure was examined using microsections according to EN ISO 17639. The intergranular corrosion resistance test was carried out according to EN ISO 3651-1 method 2.

Non-destructive testing of welded joints was carried out using the radiographic method on an RPD-250SP X-ray machine (Synthesis of NDT, Saint-Petersburg, Russia) with a control precision of 200 μm in compliance with the requirements for a welded joint under ISO 13919-1 (quality level B).

Two specimens were welded for subsequent testing in a certified laboratory to qualify welding and cladding procedures under ISO 15614-11 and qualify fusion welding operators under BS EN ISO 14732. Samples after recovery cladding were subjected to penetrant test under ISO 3452-1 and radiographic inspection under ISO 17636-1. In compliance with ISO 9712, certified specialists of at least 2nd level carried out non-destructive testing. The results of non-destructive testing were documented and confirmed by the protocol.

### 2.4. SEM Microstructure and Chemical Composition

The Scanning Electron Microscope (SEM) Carl Zeiss Auriga Crossbeam (Carl Zeiss, Jena, Germany) was used for a deeper study of the microstructure and point chemical analysis. Studies of the microstructure and chemical composition of various sections of the weld before and after heat treatment were carried out. To study the morphology and microstructure of the samples, the SE (secondary electron) regime was used. X-ray spectral analysis based on the Oxford Instruments INCA X-Max energy dispersive spectrometer (EDS) (resolution 127 eV, Carl Zeiss, Jena, Germany) the determination error was 0.05% was used for investigating the chemical composition of the samples. The study included the distribution analysis of chemical elements in various sections of the weld seam—fusion lines, various passes, sections of different crystallization (AF-mode, FA-mode), and various inclusions between grains. For spectra construction current, 700 pA and accelerating voltage 20 kV was used.

## 3. Results and Discussion

### 3.1. The Influence of the Welding Parameters on Bead-on-Plate Test

Figure 4 shows photographs of some cladded beads after a bead-on-plate test, which show that the main variable parameters significantly affect the bead width, filler wire fusibility, weld pool shielding, and the presence of a crater at the end of the weld.

Figure 5 shows photographs of the macrostructure of the bead-on-plate test obtained according to the modes presented in Table 3. They show the depth of penetration into the substrate, the bead height, the bead width, the effect of the laser beam wobbling on the melting of the substrate, the substrate heat-affected zone, the presence of pores, and lack of fusion. This indicates a significant influence of the variable parameters on the shape of the cladded bead and the resulting defects.

During the experiments, it has been found that it is difficult to obtain a weld bead with a width of more than 3 mm without defects (samples No. 1, 2). An increase in power led to very high substrate penetration (more than 2–3 mm), the formation of defects in the form of pores, and a lack of fusion.

Based on this, it was decided to use a laser beam wobbling system, which allows the beam to oscillate in a plane perpendicular to the direction of cladding. The oscillation amplitude was set in the range 1.3–2.3 mm. The oscillation frequency was 100 Hz. A further increase in the amplitude leads to a significant increase in the width of the cladded bead and the formation of a lack of fusion (sample No. 35).

Samples No. 5, 8, 9, 10, 12, 13, and 14 were obtained using the amplitude of the laser beam wobbling of 1.3 mm; their fusion line has a curved shape with two peaks (highlighted in yellow), the distance between the peaks is 1.1–1.3 mm. Samples No. 32, 33, and 34 were obtained using a larger amplitude of the laser beam wobbling (2.3 mm); their fusion line has a curved shape with two peaks (highlighted in blue) with a significantly larger distance between the peaks—2.0–2.4 mm. For samples No. 4, 30, 31 the fusion line is almost uniform (highlighted in green), the peaks are not noticeable. Samples No. 27, 28, and 29 have a uniform transition line (highlighted in orange).

Based on the results of the experimental study, a laser cladding mode with the highest quality of formation of the bead (sample No. 13, Figure 6) was established. For multipass cladding, mode No. 13 was selected; the bead height from the substrate was 1672 µm, the width was 3939 µm, and the average penetration depth into the substrate was 776 µm.

### 3.2. Multipass Laser Recovery Cladding

Figure 7 shows photographs of the macrostructure of a multi-pass cladding; sample 1 was obtained using two passes, the rest using three passes. Sample 1, obtained using two passes, has a defect in the form of a pore (200 μm in diameter) in the center of the lower part of the second pass and a slight lack of fusion on the fusion line of the second pass and the base metal. Sample 3 also exhibits defects between the first and second passes, smaller pores (80–120 µm), and substantially larger lack of fusion (600 µm). An insignificant defect is observed in sample 5, a pore on the fusion line and the second pass (~80 μm). Samples 2 and 4 have no visible defects. The geometric dimensions of the deposited metal also differ; sample 4 has the maximum height of the top bead (0.88 mm), samples 1, 2, and 3—width (3.7 mm), sample 3—depth (6.1 mm). The smallest height of the third pass bead is good for minimizing subsequent machining. The reasons for the appearance of defects can be non-melting of the wire, unstable hydrodynamic behavior of the melt pool caused by the laser beam wobbling. A detailed study of the microstructure between the passes and in the sections of the pass and fusion line is presented below.

### 3.3. Non-Destructive Control

The samples obtained after multi-pass cladding were subjected to radiographic control under GOST 7512-82. Figure 8 shows the appearance and X-ray of the obtained samples.

Digital images of radiographs were quantitatively analyzed. Using the Digimiser software, the linear dimensions of the internal defects of the weld seam, namely, pores and slag inclusions, were obtained. Next, the total area of the processed metal defects projection was calculated, and the area ratio of the deposited metal defects projection to the cladded metal projection was obtained. According to ISO 13919-1, this ratio should be less than 0.7%.

For this optimal mode, the area ratio of the weld seam defects projection to the weld seam projection was 0.17%. Table 5 presents the detailed calculation.

### 3.4. Microstructure and Microhardness

At first glance, the microstructure of the weld seam metal is the classic microstructure of a weld seam from 316L steel, obtained by laser welding, found among other authors [1,10,11] (Figure 9). All passes have a different crystallization direction; the higher the pass number, the more the dendrites are directed upward. In the first and second passes, the dendrites are directed mainly towards the center; in the third pass, the crystallization direction is toward the center and upward. This difference is most likely associated with different cooling rates and surface tension forces of molten metal in a shielding gas atmosphere. The shielding gas in a narrow groove creates more pressure and cools the molten pool less than on the surface when filling the third pass when the gas diverges in different directions and cools the weld metal more efficiently.

Figure 9 shows that the metal of the fusion line does not differ significantly depending on the heat treatment. The cladded metal of the first, second, and third passes differs only in the crystallization direction.

The microstructure shows that the FA crystallization mode matches the one calculated by formula (3). In all three passes, crystallization is observed according to the FA mode, while in the middle of the second pass of the NHT sample there is a region (length is 600–700 μm, width is 100–150 μm, Figure 10 second pass) crystallized according to the AF mode, which is due to the larger volumes of the melt pool and, accordingly, lower cooling rates, especially in the central part of the second pass, which is consistent with the results of [9]. This crystallization mode can lead to the formation of crystallization cracks; however, in our case, no cracks were observed, even after mechanical tests. A detailed study of the samples using SEM on the HT sample also revealed the same area of slightly smaller dimensions.

The metal microstructure of the corresponding sections of the NHT and HT samples does not differ significantly depending on the heat treatment. Until the middle of the third pass, columnar grains are observed directed towards the bead center. In all areas, a first-order dendritic structure is observed, except for the third pass top. In the middle of the third pass, equiaxed grains are observed, in the upper part of the third pass, differently oriented austenite dendrites with a large amount of ferrite are observed. This is justified by the higher cooling rates caused by the proximity to the surface and heat dissipation due to the shielding gas blowing.

Microhardness measurements were carried out in manual mode in the following sections: across the middle of each pass (green lines, Figure 9) and in the center, from the bottom to top through all three passes (red line, Figure 9). When measuring from the bottom to top along the green line, regardless of the heat treatment effect, on average, the microhardness of the third pass was 220–250 HV, and it was 20–25 HV higher than microhardness of the second pass (210–235 HV). The microhardness of the second pass was 15–20 HV higher than the microhardness of the first pass (190–220 HV). The minimum microhardness was observed in the upper part of the third pass as 190 HV, the maximum in the metal of the first pass, before reaching the fusion line, measuring 250 HV. This difference is explained by the absence of heating before the first pass and the presence of heating after the first and second passes (interpass temperature was 150 °C). Additionally, it is explained by the formation in the upper part of the third passage of a large amount of δ-ferrite, which is a more plastic phase compared to austenite.

Heat treatment at a temperature of 450 °C did not significantly affect the microhardness values. It should be noted that heat treatment of austenitic steels does not significantly affect the change in microhardness the changes occur in the range of 20–40 HV, while the microhardness of carbon steels can change by several times. Heat treatment significantly affects the performance properties of austenitic steels (corrosion resistance and cyclic strength). When measuring the microhardness of the weld seam metal for each pass of the NHT (Figure 11a) and HT (Figure 11b) samples, no significant differences were observed. The measured values are in the range of 200–230 HV, regardless of the pass number and the heat treatment mode. Exceptions are measurements made near the fusion line on the base metal (245 HV). As mentioned earlier, when measuring from the bottom to the top through all three passes, the minimum microhardness is observed at the third pass of the NHT sample (190 HV).

### 3.5. SEM

Due to the fact that optical microscopy has limitations on magnification, SEM studies were carried out, including the distribution analysis of chemical elements in various sections of the weld seam—fusion lines, various passes, sections of different crystallization (AF-mode and FA-mode), and various inclusions between the grains.

Figure 12 shows the microstructure of the central part of the second pass, on which the presence of two crystallization modes was observed—the FA-mode on the right and left parts, and AF-mode in the central part, which was also observed in the study on OM (Section 3.4). The studies were carried out at a magnification of 350, 1000, 3500, and 7500. In view of the fact that the microstructure of similar welded joints and deposited metal has been sufficiently studied in other works [1,2,3], in this work, the main attention was paid to the chemical composition analysis in different areas: (1) in the FA-mode and AF-mode sections; and (2) in the dark (Figure 13, red arrows) and the light zones of these sections (Figure 13, yellow arrows). As already shown in other works on the point chemical composition in different areas of the weld seam, it is possible to calculate *Cr_eq_/Ni_eq_* [41], which can differ significantly, and, accordingly, the crystallization mode too. Additionally, for high cooling rates inherent in laser welding, the *Cr_eq_/Ni_eq_* ratio changes from 1.50 to 1.65 for the crack resistance not to appear.

Comparative analysis of the data on the chemical composition of the NHT sample in the areas of AF-crystallization and FA-crystallization (Figure 12) shows that these areas do not differ significantly in chemical composition. Except for the nickel content, which is less in the light FA area (15.83 Ni) than in the light AF area (18.12 Ni), the calculations were performed separately (the table in Figure 12 shows averaged values). There is also Mo, which is more in the dark AF area (3.44 Mo) than in the light FA area (2.04 Mo). Meanwhile, for the HT sample after heat treatment, a difference in the nickel content is also observed, but not so significant; nickel content in the light FA area (13.34 Ni) is less than that in the light AF area (12.58 Ni). In general, the HT sample after heat treatment has a more uniform distribution of alloying elements in the light and dark areas compared to the NHT sample, except for molybdenum, the content of which in the light and dark areas differs by 40%. This shows the effect of heat treatment on the coagulation process, and the formation of easily precipitates brittle Mo-rich phases.

This assumption is confirmed by a more detailed study of the Mo distribution. Figure 13a shows the photograph of the metal in the middle of the second pass crystallized by FA-mode (magnification 1500), which shows precipitate phases formed along grain boundaries after heat treatment. Figure 13b shows a similar area at higher magnification (magnification 7500) in which these particles are seen much better. Point chemical analysis showed that there is about 5.50% Mo in these particles.

The study of the distribution of the chemical elements was carried out as the studies on the OM showed that the weld seam metal of the third pass differs significantly from the first and second pass. The area of 30–40 microns to the right of the fusion line was selected (Figure 14). A comparison of the chemical composition was also carried out in light and dark areas. Studies show that the HT sample after heat treatment has almost uniform distribution in the light and dark areas compared to the sample without NHT. In the NHT sample, the difference in the chemical composition in the light and dark areas is Si—50%, Cr—20%, Ni—50%, and Mo—60%. This is justified by the high cooling rates in this section and the minimum time for diffusion processes. Sections with possible precipitate phases of the HT sample were not found in this section of the seam.

When comparing the dark and light areas of AF and FA crystallization in the sample before (NHT) and after (HT) heat treatment (Figure 12), it was revealed that the chemical composition in the corresponding areas also changed. As Cr and Mo are ferrite-forming elements, they easily dissolve in ferrite. The content of Mo in the dark area in the HT sample is 1.48% less than in the NHT sample, which is probably due to the more uniform dissolution of molybdenum in ferrite after heat treatment. Conversely, the Mo content in the light area before and after heat treatment in the NHT sample is 1.22% less than in the HT sample, which, on the contrary, is caused by a lower dissolution of Mo in austenite. Comparing the chromium content in the dark area of the NHT and HT samples, it can be seen that the difference is insignificant (20.80% and 19.87%, respectively), just as in the light area in the NHT and HT samples (19.46% and 18.81%, respectively). After heat treatment, the amount of chromium decreased by about 0.9% in both the dark and light areas. It is probably justified by, firstly, a sufficiently large amount of chromium, and secondly, its less active migration in the weld seam metal after welding and after heat treatment.

After heat treatment, the amount of nickel decreased in both the dark and light areas. After welding (sample NHT), the amount of nickel in the studied areas was significantly higher (16.97—Light area, 13.00—Dark area) than in the base metal (about 12%). After heat treatment (sample HT), the amount of nickel in the weld seam metal significantly decreased (12.96—Light area, 10.80—Dark area) and, on average, was the norm for this alloy (about 12%). In the light area, which is austenite, there is naturally more nickel, almost 13%, which is an austenite-forming element, compared to the dark area, in which there is almost 3% less nickel.

## 4. Conclusions

The paper presents the study results of the influence of the modes, in particular the laser beam wobbling, during three-pass laser cladding in a narrow slot groove of 316L steel, without beveling the edges.

Based on the results of experimental studies, the technology of laser welding with filler wire with the formation of welded joints that meet the ISO 13919 standard (quality level B) was developed. The developed technology of laser welding with filler wire has successfully passed the certification for conformity with the ISO 15614-11 standard.The optimal modes of laser beam wobbling were selected; the amplitude was 1.3 mm, the frequency 100 Hz, at which the bead parameters were height—1672 µm, width—3939 µm, and the depth of penetration into the substrate—776 µm.Comparative studies of microhardness and microstructure of different passes are consistent, and they show that the lower part of the first pass has the maximum microhardness (250 HV), and the upper part of the third pass has the minimum (190 HV). This correlates with the microstructure; in the upper part of the third pass, there is significantly more δ-ferrite, which is a more plastic phase than austenite. Additionally, in the upper part of the third pass, austenite has a differently oriented dendritic structure, and in the remaining passes (1 and 2), a columnar and fine granular structure with a lower content of δ-ferrite.Heat treatment has influenced the reduction in thermal stresses. It had no significant effect on the change in microstructure and microhardness. However, it significantly influenced the more uniform distribution of chemical elements in the HT sample. Additionally, the point chemical analysis in the central part of the second pass of the HT sample along the grain boundaries revealed particles with a significantly higher Mo content (5.50%), presumably precipitated phases [8]. The following works could continue the study of the heat treatment effect on the coagulation of these phases.Comparison of the chemical elements’ distribution between the second and third passes for the HT sample in the corresponding areas have approximately the same values; the deviations for all points are about 1%. Meanwhile, the NHT sample significantly differs in the content of ferrite-forming elements (Cr—24.52% and Mo—5.08%) in the dark area, which is consistent with the microstructure analysis and microhardness measurements.

## Figures and Tables

**Figure 1 materials-15-00722-f001:**
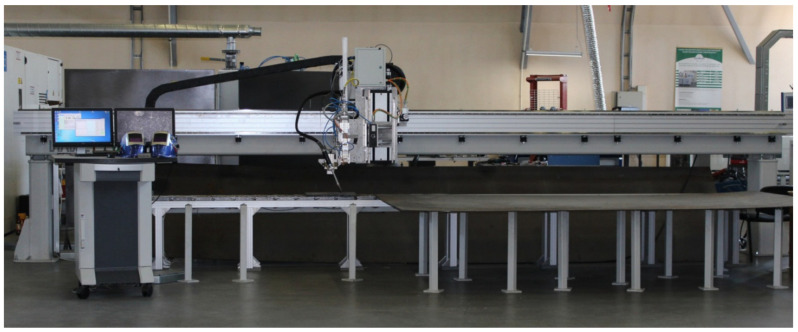
External view of the technological laser complex.

**Figure 2 materials-15-00722-f002:**
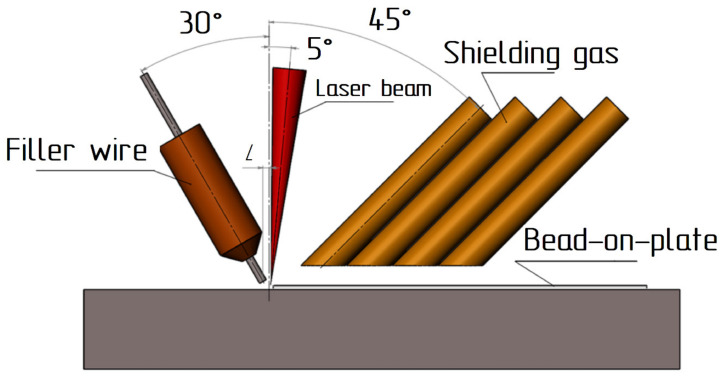
Scheme of laser cladding with filler wire.

**Figure 3 materials-15-00722-f003:**
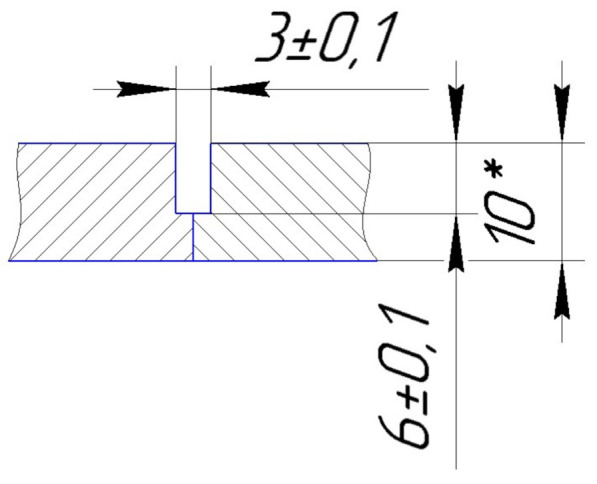
Geometric dimensions of the samples before welding.

**Figure 4 materials-15-00722-f004:**
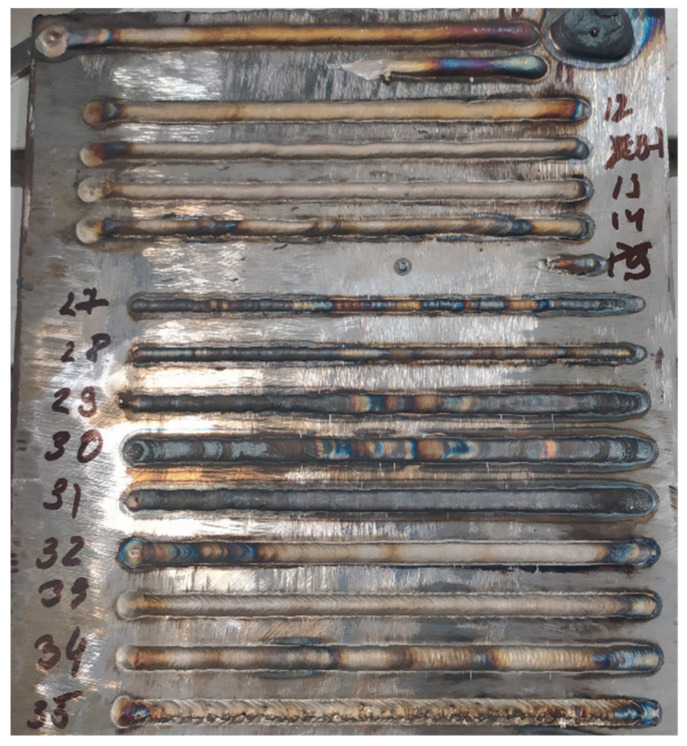
Photo of bead-on-plate test.

**Figure 5 materials-15-00722-f005:**
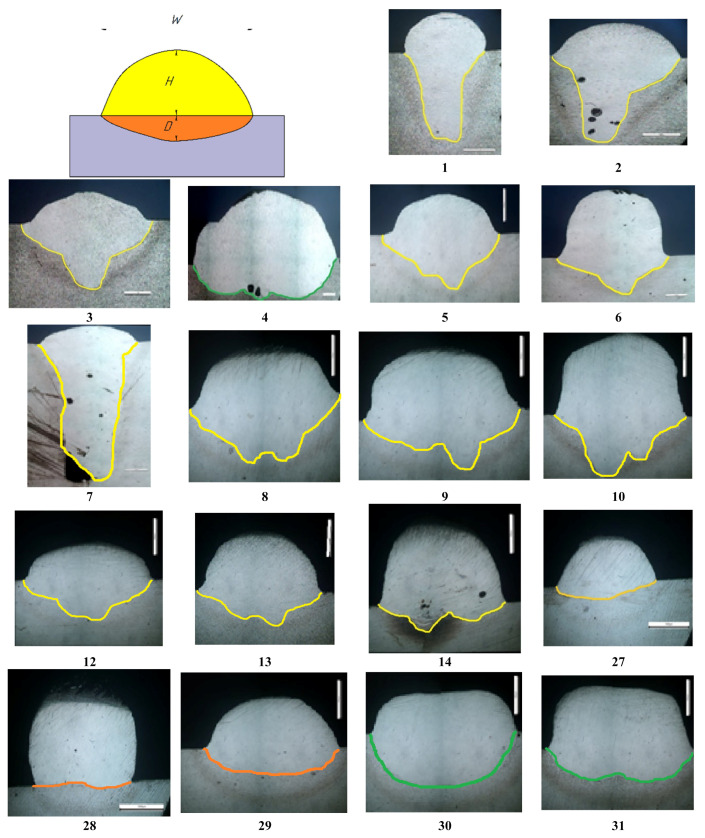
Macrostructure of samples No. 1–35.

**Figure 6 materials-15-00722-f006:**
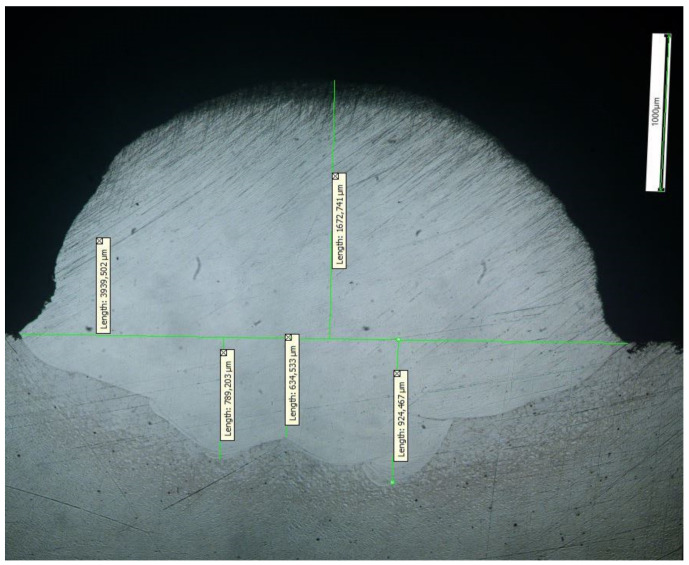
Macrostructure of the bead obtained by mode No. 13.

**Figure 7 materials-15-00722-f007:**
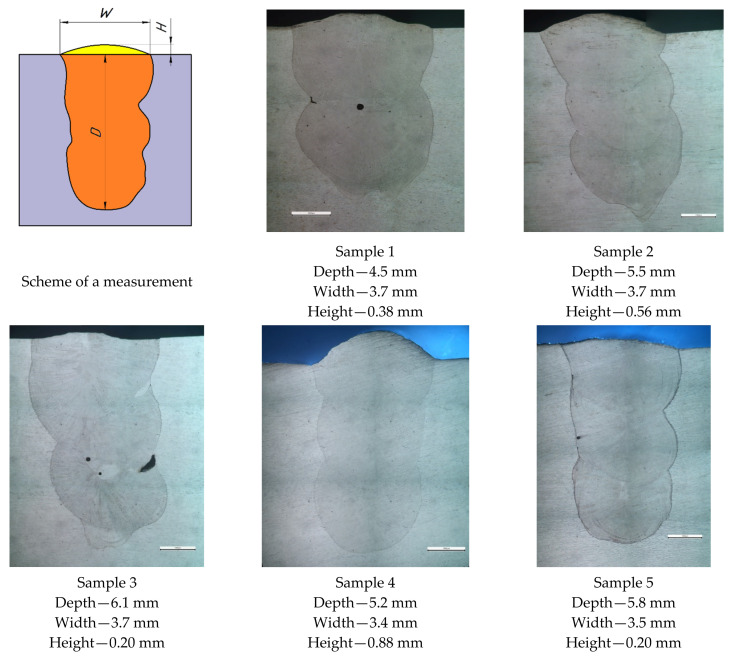
Macrostructure of multi-pass cladding samples obtained by mode 13.

**Figure 8 materials-15-00722-f008:**
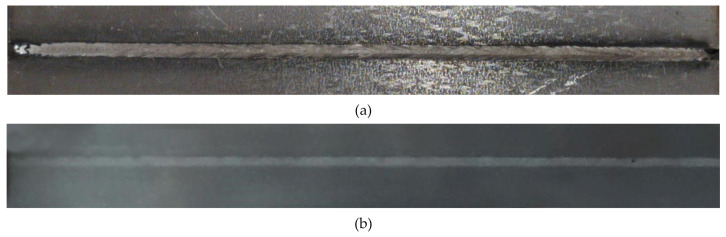
Appearance (**a**) and X-ray (**b**) of the sample front.

**Figure 9 materials-15-00722-f009:**
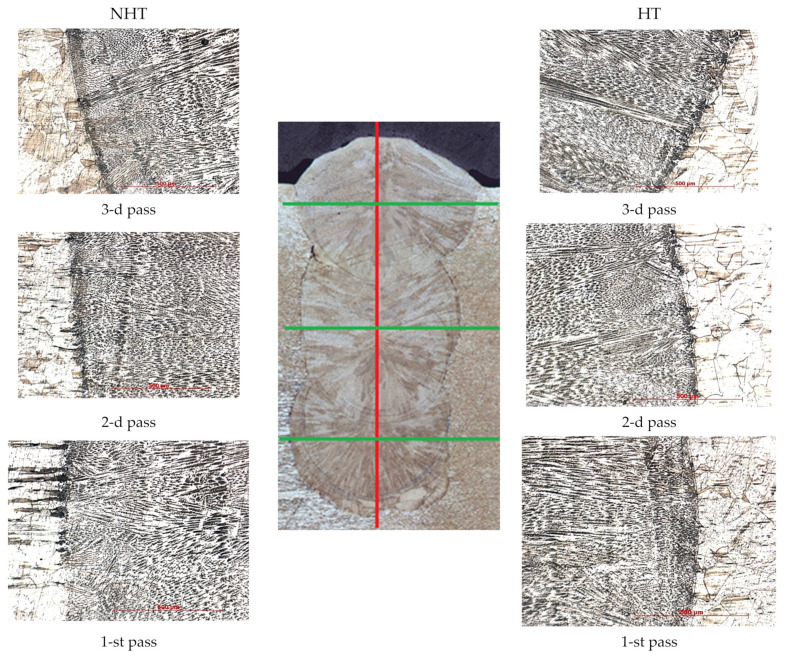
The weld seam macrostructure is in the **center**; the transition line from the base metal to the middle of 1, 2, and 3 passes of NHT sample—on the **left**; and the transition line from the base metal to the middle of 1, 2, and 3 passes of HT sample—on the **right**.

**Figure 10 materials-15-00722-f010:**
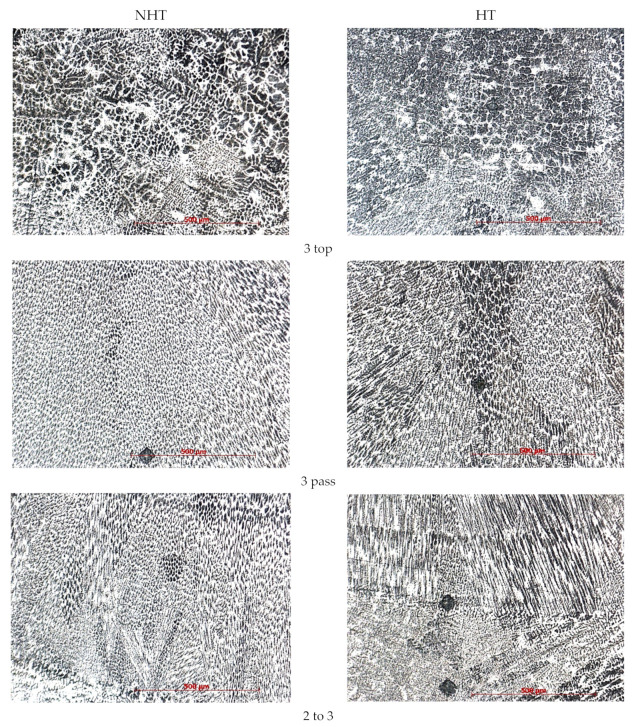
Microstructure of the central part of the cladded metal of different parts of the sample (bottom, first pass, transition line of first pass to second, second pass, transition line of second pass to third, third pass, and top of third pass) × 200.

**Figure 11 materials-15-00722-f011:**
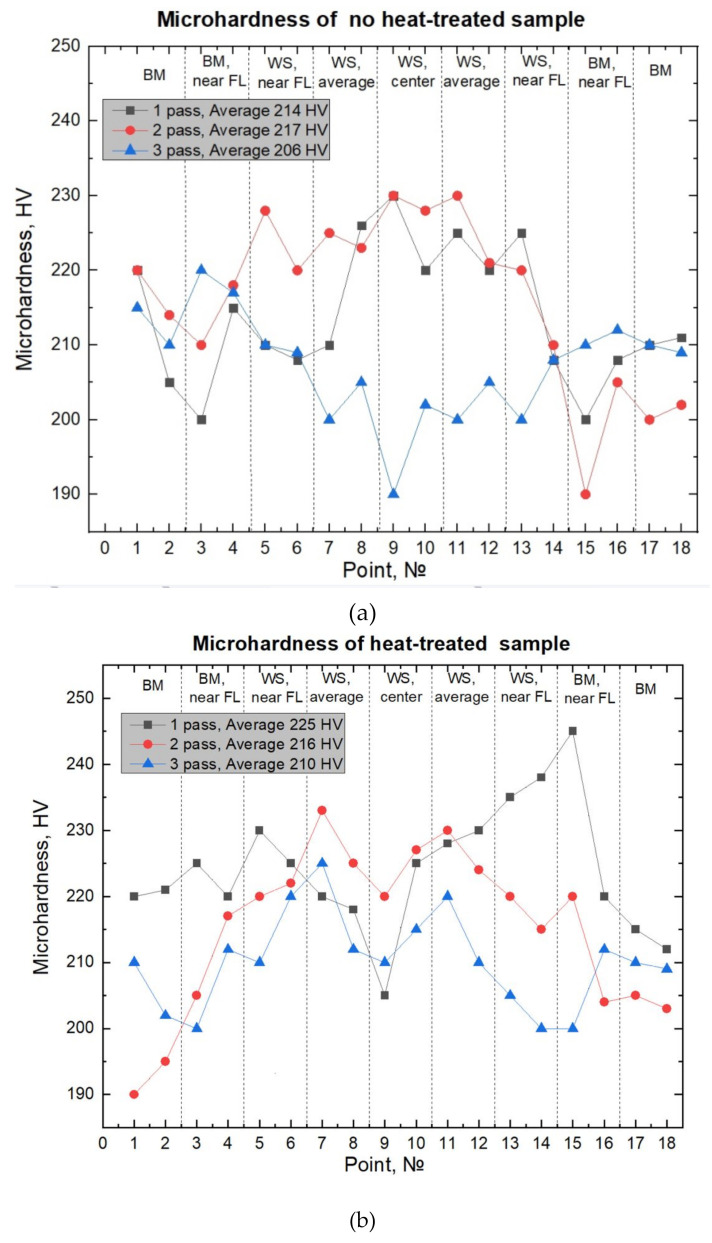
Microhardness of (**a**) NHT and (**b**) HT samples.

**Figure 12 materials-15-00722-f012:**
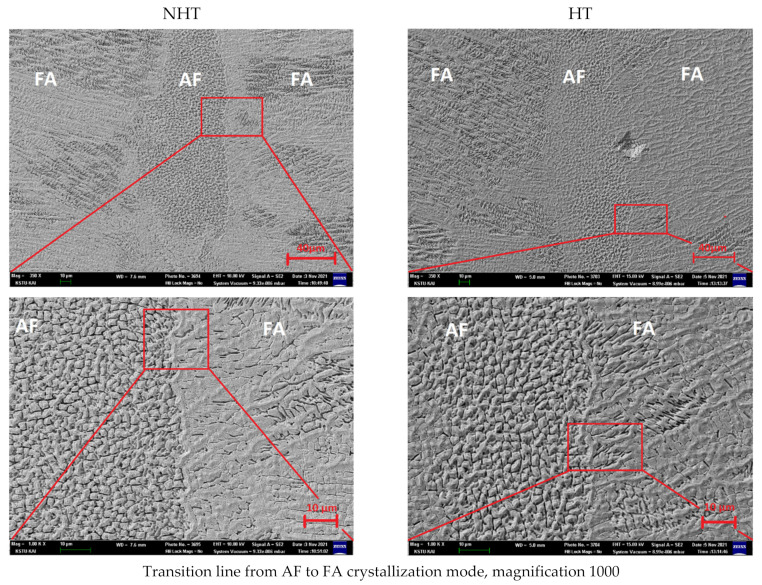
Microstructure and distribution of chemical elements of the middle part of the second pass (left NHT, right HT, magnification 350, 1000, and 3500).

**Figure 13 materials-15-00722-f013:**
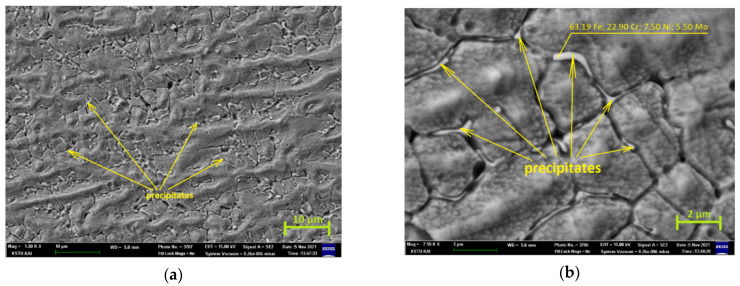
Microstructure of the middle of the second pass of the HT sample, magnification (**a**) 1500 and (**b**) 7500.

**Figure 14 materials-15-00722-f014:**
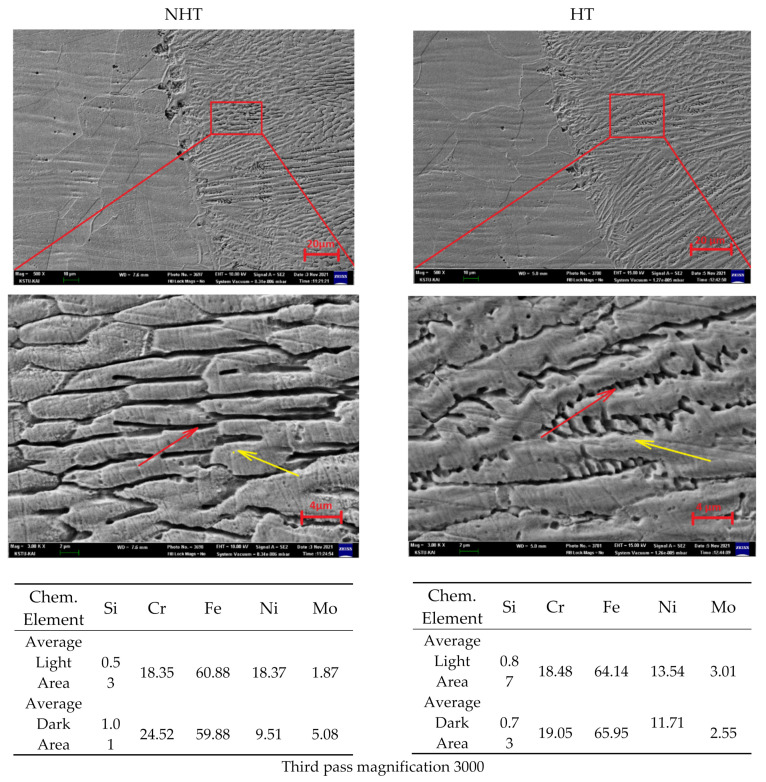
Microstructure and distribution of chemical elements of the upper part of the third pass (left NHT, right HT, magnification 500 and 3000).

**Table 1 materials-15-00722-t001:** Technical characteristics of the technological complex.

No.	Parameter, Unit of Measurement	Value
1	Maximum size of processed rolled products, mm	12 × 3200 × 7000
2	Maximum power of the laser source, kW	16
3	Maximum linear speed, m/min	4
4	Adjustment range of linear movements speed, m/min	from 0.4 to 4
5	Maximum speed of idle movements, m/min	10
6	Limit of the relative error of speed maintaining, %	±5
7	Maximum deviation from straightness when moving, mm	±0.5

**Table 2 materials-15-00722-t002:** Chemical composition of the base and filler material.

Welding Materials	C	Si	Mn	Ni	Cr	Mo	Fe	S	P
Base metal	0.03	0.75	2	12	16	2.5–3	~67	0.030	0.045
Filler metal	0.02	0.41	2	12	19	2.5–3	~64	0.030	0.045

**Table 3 materials-15-00722-t003:** Variable laser cladding parameters.

No.	P, W	WS, mm/s	FWS, m/min	Amplitude, mm
1	4470	15	2.4	No wobbling
2	4470	15	2.4	No wobbling
3	4470	15	2.4	1.3
4	4470	1.8	4.9	1.3
5	4470	15	4.9	1.3
8	4470	15	4	1.3
9	4470	15	4.5	1.3
10	4470	15	5	1.3
12	2556	10	2.5	1.3
13	2556	10	2.4	1.3
14	2556	10	4.9	1.3
27	1530	5	1.5	1.3
28	1530	5	2	1.3
29	1940	5	2	1.3
30	2300	5	2	1.3
31	2300	5	2.5	1.3
32	2556	10	2.5	2.13
33	2556	10	2.4	2.13
34	2300	5	2.5	2.13
35	2556	10	2.5	3.2

**Table 4 materials-15-00722-t004:** Parameters of the optimal mode of laser cladding with filler wire.

Parameters	Value
Laser irradiation Power, W	2550
Welding speed, mm/s	10
Filler wire diameter, mm	1
Shielding gas flow rate, L/min	20
Deepening the focus relative to the product surface, mm	12
Welding head inclination angle, deg.	0
Filler wire feed speed, m/min	4.2
Filler wire feed angle relative to the laser beam, deg.	30
Amplitude,%	50
Frequency, Hz	100

**Table 5 materials-15-00722-t005:** Calculation of the area ratio of defects to the seam.

Width, mm	3			
Length, mm	200			
Area, mm^2^	600			
Defect number	Diameter, mm	Length, mm	Width, mm	Area, mm^2^
Defect 1	0.2			0.003
Defect 2	0.4			0.13
Defect 3	0.8			0.5
Defect 4	0.7			0.38
Total area of defects, mm^2^	1.04			
Area ratio of defects to seam,%	0.17			

## Data Availability

Data sharing is not applicable for this article.

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
