# Peer review of "Influence of Laser Beam Wobbling Parameters on Microstructure and Properties of 316L Stainless Steel Multi Passed Repaired Parts"

_materials, 2022, doi:10.3390/ma15030722_

Round 1
Reviewer 1 Report
The authors have presented the results of optimizing the repair of a model samples using the multi pass laser cladding with filler wire. The optimization process has been presented in a comprehensive manner and the results are sound. The reviewer has only a few minor corrections or suggestions.
- The first sentence in abstract reads well if you delete "using the technology of"
- The second sentence in abstract reads well if you delete "The study results of"
- In line 45 just before stress corrosion cracking add "undergo"
- Lines 67-68 seems to be written in Russian. Please delete.
- In equation 2 "eq" on Ni should be in subscript.
- Add references for the statement in line 73-74 as well as 77-79.
- Line 140 add full stop.
- Line 185 - How was the chemical composition determined? For some elements a range is given. What is in general the error on the composition?
- For SEM images the magnification is given but what are the length scales? In optical mages the 500 micrometer length scale is given. It is difficult to understand the SEM images without these scales. May be it is there in the black strip underneath each image. But it is blurred and cannot be seen.
- What are the error bar on the elementary composition determined using EDS?
- It is argued that the appreciable elemental composition difference observed in NHT samples at the pits and on grains is due to insufficient diffusion because of high cooling rates. But the migration energies for each atom are widely different. Element/s with highest migration should be coagulating. How is that all elements including Fe have such large variation?
- What is the variance on the micro hardness measured at each position? The plot has a suppressed zero and because of these the variations can be thought to be realistic. However if you repeated the measurement at position 1 say 5 times what is the average and variance.
Author Response
The authors thank the distinguished reviewer for the work done, the positive assessment of the manuscript, and are confident that these comments will significantly improve the quality of this article.
For the convenience of reviewers, corrections are highlighted in different colors:
Reviewer 1
Reviewer 2
Comments and Suggestions for Authors
The authors have presented the results of optimizing the repair of a model samples using the multi pass laser cladding with filler wire. The optimization process has been presented in a comprehensive manner and the results are sound. The reviewer has only a few minor corrections or suggestions.
- The first sentence in abstract reads well if you delete "using the technology of"
The comment taken into account; the corresponding changes have been made.
- The second sentence in abstract reads well if you delete "The study results of"
The comment taken into account; the corresponding changes have been made.
- In line 45 just before stress corrosion cracking add "undergo"
The comment taken into account; the corresponding changes have been made.
- Lines 67-68 seems to be written in Russian. Please delete.
The comment taken into account; this sentence has been deleted.
- In equation 2 "eq" on Ni should be in subscript.
In equation 2 "eq" on Ni has been moved to subscript.
- Add references for the statement in line 73-74 as well as 77-79.
References to Lippold and Kovachevich added [1, 3].
- Line 140 add full stop.
Point added. Thank you for your attention!
- Line 185 - How was the chemical composition determined? For some elements a range is given. What is in general the error on the composition?
The specified range was taken from the certificate for the supplied base metal and filler wire. The content of chromium and nickel is indicated in a narrower range.
- For SEM images the magnification is given but what are the length scales? In optical mages the 500 micrometer length scale is given. It is difficult to understand the SEM images without these scales. May be it is there in the black strip underneath each image. But it is blurred and cannot be seen.
Scale bars added on all SEM photos, Figures 12, 13, 14.
- What are the error bar on the elementary composition determined using EDS?
The error in determining the chemical composition using EDS depends on many factors and methods and constitute 0.05%. Data added in section 2.5. SEM Microstructure and Chemical Composition.
- It is argued that the appreciable elemental composition difference observed in NHT samples at the pits and on grains is due to insufficient diffusion because of high cooling rates. But the migration energies for each atom are widely different. Element/s with highest migration should be coagulating. How is that all elements including Fe have such large variation?
The reviewer is absolutely right; the rate of migration of various chemical elements is significantly different. Due to the fact that the done heat treatment is not high-temperature and the main aim is to relieve stress, migration and segregation not of all elements is the same.
This manuscript presents an analysis of the distribution of chemical elements in various areas with AF-crystallization and FA-crystallization; dark and light areas of the AF and FA crystallization. But a comparative analysis of dark and light areas of the AF and FA crystallization before and after heat treatment is not presented, which just shows how the content of chemical elements differs in specific areas.
Accordingly, the following text has been added to the manuscript:
When comparing the dark and light areas of AF and FA crystallization in the sample before (NHT) and after (HT) heat treatment, it was revealed that the chemical composition in the corresponding areas also changed. Since Cr and Mo are ferrite-forming elements, they easily dissolve in ferrite. The content of Mo in the dark area in the HT sample is 1.48% less than in the NHT sample, which is probably due to the more uniform dissolution of molybdenum in ferrite after heat treatment. Conversely, the Mo content in the light area before and after heat treatment in the NHT sample is 1.22% less than in the HT sample, which, on the contrary, is caused by a lower dissolution of Mo in austenite. Comparing the chromium content in the dark area of the NHT and HT samples, it can be seen that the difference is insignificant (20.80%) and (19.87%), respectively, just as in the light area in the NHT and HT samples, it can be seen that the difference is insignificant (19.46% ) and (18.81%) respectively. After heat treatment, the amount of chromium decreased by about 0.9% in both the dark and light areas. It is probably justified by, firstly, a sufficiently large amount of chromium, and secondly, its less active migration in the weld seam metal after welding and after heat treatment.
After heat treatment, the amount of nickel decreased in both the dark and light areas. After welding (sample NHT), the amount of nickel in the studied areas was significantly higher (16.97 - Light area, 13.00 - Dark area) than in the base metal (about 12%). After heat treatment (sample HT), the amount of nickel in the weld seam metal significantly decreased (12.96 - Light area, 10.80 - Dark area) and, on average, was the norm for this alloy (about 12%). In the light area, which is austenite, there is naturally more nickel almost 13%, which is an austenite-forming element, compared to the dark area, in which there is almost 3% less nickel.
- What is the variance on the micro hardness measured at each position? The plot has a suppressed zero and because of these the variations can be thought to be realistic. However if you repeated the measurement at position 1 say 5 times what is the average and variance.
Figure 11 is redone, some clarifications have been added to the text.
Heat treatment at a temperature of 450 0C did not significantly affect the microhardness values. It should be noted that heat treatment of austenitic steels does not significantly affect the change in microhardness the changes occur in the range of 20 - 40 HV, while the microhardness of carbon steels can change by several times. Heat treatment significantly affects the performance properties of austenitic steels (corrosion resistance, cyclic strength). When measuring the microhardness of the weld seam metal for each pass of the NHT (Figure 11 a) and HT (Figure 11 b) samples, no significant differences were observed. Basically, the measured values are in the range of 200 - 230 HV, regardless of the pass number and the heat treatment mode. Exceptions are measurements made near the fusion line on the base metal (245 HV). As mentioned earlier, when measuring from the bottom to the top through all three passes, the minimum microhardness is observed at the third pass of the NHT sample (190 HV).

Reviewer 2 Report
Dear Authors,
the work is interesting, although I have some comments:
- l.67-68 - the text is not translated into English
- Table 1 - No ??
- l.185 - should be Table 2
- Check the style in the whole text, e.g., l. 188 - Before multipass cladding, one bead was cladded on a flat substrate to identify the optimal parameters of laser cladding with a cold filler wire - cladding x3. Maybe "process"?
- l. 191 Table 3 should be Table 4. Table 3 not cited.
- How the optimal set of parameters was selected? Was it based on any calculations? How the quality was estimated - the best quality of sample no 13?
- Were the test repeated?
- Figure 7 and others: the values are not readable.
- Figure 11 - not cited. It is difficult to understand presented results if it is not cited. The results are very random. Is there any tendency?
- General conclusion is that the results are interesting, but the way of presentation should be better.
Best regards,
Reviewer
Author Response
The authors thank the distinguished reviewer for the work done, the positive assessment of the manuscript, and are confident that these comments will significantly improve the quality of this article.
For the convenience of reviewers, corrections are highlighted in different colors:
Reviewer 1
Reviewer 2
Comments and Suggestions for Authors
Dear Authors,
the work is interesting, although I have some comments:
- 67-68 - the text is not translated into English
We ask our deepest apologies for such an oversight. This sentence has been removed.
- Table 1 - No ??
п.п. is deleted from Table 1.
- 185 - should be Table 2
Exactly on line 185 (old version) Table 2 should be written instead of Table 1.
- Check the style in the whole text, e.g., l. 188 - Before multipass cladding, one bead was cladded on a flat substrate to identify the optimal parameters of laser cladding with a cold filler wire - cladding x3. Maybe "process"?
I absolutely agree, the sentence has been rewritten. Changes have also been made to other similar sentences throughout the text.
“Before multipass cladding, one bead was deposited on a flat substrate to identify the optimal parameters of laser process with a cold filler wire.”
- 191 Table 3 should be Table 4. Table 3 not cited.
Table 3 and Table 4 are cited in the text and the corresponding changes have been made.
- How the optimal set of parameters was selected? Was it based on any calculations? How the quality was estimated - the best quality of sample no 13?
Quality was established for optimum performance, minimum penetration into the substrate and maximum wettability angle.
- Were the test repeated?
As indicated in Table 3, quite a few regimes were used when developing the technology of laser cladding of one bead, some of them were similar. After the analysis of the macrostructure and the choice of mode No. 13, several additional control single bead cladding were carried out, before the multi-pass process.
- Figure 7 and others: the values are not readable.
Exactly on Figure 7 the averaged values of all basic dimensions are presented under each sample. On the rest of the Figures (12, 13, 14) scale bars added.
- Figure 11 - not cited. It is difficult to understand presented results if it is not cited. The results are very random. Is there any tendency?
Figure 11 redone. The following text has been added to the manuscript.
Heat treatment at a temperature of 450 0C did not significantly affect the microhardness values. It should be noted that heat treatment of austenitic steels does not significantly affect the change in microhardness the changes occur in the range of 20 - 40 HV, while the microhardness of carbon steels can change by several times. Heat treatment significantly affects the performance properties of austenitic steels (corrosion resistance, cyclic strength). When measuring the microhardness of the weld seam metal for each pass of the NHT (Figure 11 a) and HT (Figure 11 b) samples, no significant differences were observed. Basically, the measured values are in the range of 200 - 230 HV, regardless of the pass number and the heat treatment mode. Exceptions are measurements made near the fusion line on the base metal (245 HV). As mentioned earlier, when measuring from the bottom to the top through all three passes, the minimum microhardness is observed at the third pass of the NHT sample (190 HV).
- General conclusion is that the results are interesting, but the way of presentation should be better.
The 5th point has been added to the conclusions.
Comparison of the chemical elements distribution between the second and third passes for the HT sample in the corresponding areas have approximately the same values, the deviations for all points are about 1%. Whereas the NHT sample significantly differs in the content of ferrite-forming elements (Cr - 24.52%, Mo - 5.08%) in the dark area, which is consistent with the microstructure analysis and microhardness measurements.

Round 2
Reviewer 2 Report
Dear Authors,
I accept your responses.
Best regards,
Reviewer